**Data Availability Statement:** Due to restrictions by the Makerere University School of Public Health

# Challenges and commonly used countermeasures in the implementation of lifelong antiretroviral therapy for PMTCT in Central Uganda: Health providers' perspective

**Aggrey David Mukose**[1,2]\*, **Hilde Bastiaens**[2,3], **Fredrick Makumbi**[1], **Esther Buregyeya**[4], **Rose Naigino**[5], **Joshua Musinguzi**[6], **Jean-Pierre Van Geertruyden**[2], **Rhoda K. Wanyenze**[4]

**1** Department of Epidemiology and Biostatistics, School of Public Health, College of Health Sciences, Makerere University, Kampala, Uganda, **2** Department of Epidemiology and Social Medicine, Faculty of Medicine and Health Sciences, Global Health Institute, University of Antwerp, Antwerp, Belgium, **3** Department of Family Medicine and Population Health, Faculty of Medicine and Health Sciences, University of Antwerp, Antwerp, Belgium, **4** Department of Disease Control and Environmental Health, School of Public Health, College of Health Sciences, Makerere University, Kampala, Uganda, **5** Makerere University School of Public Health, Kampala, Uganda, **6** Ministry of Health, Kampala, Uganda

\* amukose@musph.ac.ug

## Abstract

### Introduction

Uganda has implemented lifelong antiretroviral therapy for the prevention of mother-to-child HIV transmission since September 2012. Implementation of this strategy has been met with health provider and client challenges which have persisted up to date. This study explored providers' perspectives on the challenges and countermeasures of the implementation and scale-up of lifelong ART among pregnant and breastfeeding women.

### Methods

A qualitative descriptive study was conducted whereby 54 purposively selected participants from six facilities in three districts of Central Uganda namely; Masaka, Mityana, and Luwero were recruited. A key informant interview guide was used to collect data from the study participants. The data were thematically analysed using Atlas-ti, Version 7.

### Results

Study participants reported challenges under the themes of 1) inadequacy of HIV service delivery (lack of relevant training, health provider shortages, inadequate counselling, stockouts of essential HIV commodities); 2) Non-utilization of HIV services (Non-disclosure of HIV-positive results, denial of HIV positive results, fear to be followed up, unwillingness to be referred, large catchment area, lack of transport); and 3) Suboptimal treatment adherence (fear of ART side effects, preference for traditional medicines, low male partner involvement in care and treatment). Strategies such as on-job training, mentorship, task shifting, redistribution of HIV commodities across facilities, accompanying of women to

Higher Degrees, Research and Ethics Committee, some access restrictions apply to the data for reasons of safety and protection of study subjects and their institutions. Sensitive qualitative data was collected from study participants and they didn't consent to open data access. However, criteria eligible researchers with interest in the data may request for anonymized data access through the Chair Higher Degrees, Research and Ethics Committee (contact via sphrecadmin@musph.ac. ug and cc:jkagaayi@musph.ac.ug).

**Funding:** This study was funded by the Global Fund through the Ministry of Health-Uganda [Grant Number: UGD-708-G07-H]. The funders had no role in study design, data collection and analysis, decision to publish, or preparation of the manuscript.

**Competing interests:** The authors have declared that no competing interests exist.

**Abbreviations:** A.M., Aggrey David Mukose; E.B., Esther Buregyeya; H.B., Hilde Bastiaens; R.N, Rose Naigino; R.W., Rhoda K. Wanyenze.

mother-baby care points, ongoing counseling of women, peers, and family support groups were commonly used countermeasures.

## Conclusions

This study highlights key challenges that health providers face in implementing lifelong anti-retroviral therapy services among pregnant and postpartum women. Context-specific, innovative, and multilevel system interventions are required at national, district, health facility, community and individual levels to scale up and sustain the lifelong antiretroviral therapy strategy among pregnant and breastfeeding women.

## Introduction

Adoption of lifelong antiretroviral therapy (ART) for all pregnant and breastfeeding women living with HIV regardless of CD4 count or clinical stage (Option B+) for prevention of mother-to-child transmission (PMTCT) of HIV started in Malawi in 2011 [1]. Subsequently, this strategy was adopted by many countries in sub-Saharan Africa (SSA), including Uganda [2–4]. At that time, this choice was based on programmatic and operational reasons, particularly in generalized epidemics where there are high fertility rates, small birth intervals, poor access to CD4 testing, and a long duration of breastfeeding [5]. Evidence suggests that initiating ART in all pregnant and breastfeeding women would reduce HIV incidence and prevent HIV transmission in both current and future pregnancies [5].

Uganda adopted the lifelong ART strategy for pregnant and breastfeeding women in September 2012. By August 2013, the strategy had been scaled up in all the 112 districts across the country [6]. Moreover, Uganda started implementing the HIV "test and treat" policy for all children, pregnant and breastfeeding women, HIV-positive people with both TB or Hepatitis B co-infection and the HIV positive individuals in sero-discordant relationships in 2014 [7]. Besides, in the same year, Uganda adopted the World Health Organization's recommendation for routine viral load testing as the standard of care compared to CD4 count for monitoring ART effectiveness [8]. However, in 2016, Uganda revised her "Consolidated guidelines for the prevention and treatment of HIV and AIDS" to include all adolescents and adults living with HIV on the "test and treat" policy. This policy involves providing lifelong ART to people living with HIV irrespective of CD4 count or World Health Organization (WHO) clinical stage [9]. These changes drastically increased the number of people living with HIV (PLHIV) who were initiated on ART which could end up constraining the health system [10]. Finally, in 2020, the Uganda Ministry of Health (MOH) rolled out revised HIV and AIDS prevention, and treatment guidelines which recommend the optimization of ART using Dolutegravir-based regimens as the preferred first line for all eligible PLHIV including pregnant and breastfeeding adolescent girls and women [11]. Typically, pregnant and lactating women are categorized under priority populations since they have a higher chance of acquiring or transmitting HIV. Besides, HIV-infected pregnant and postpartum women have unique needs including access to and utilization of HIV services, retention in care and adherence [12, 13]. Consequently, the WHO anticipated that additional support would be required to ensure optimal adherence and retention in HIV care for HIV positive women initiating lifelong ART since many would still be healthy [5]. As a result, mother-to-child transmission (MTCT) of HIV rate would rapidly abate. Although the MTCT rates in Uganda have declined over time, the desired level has not been achieved. For instance, a recent (2017–2019) impact evaluation study found that the

overall MTCT rate in Uganda at 18 months post-delivery was 2.8% (95% CI: 2.0–3.9) [14] while an annual performance report showed that an estimated 5,300 MTCT infections occurred in the 2020/21 financial year [15]. Ergo, Uganda still lags behind in attaining the validation status for elimination of mother to child transmission (e-MTCT) of HIV.

To maximise the benefits of lifelong ART among pregnant and breastfeeding women, there is thus a need for a fully functional and organized health system to support the women and their infants. However, some studies have identified health system challenges in PMTCT implementation. Some of the challenges include; patient and health system difficulties such as poor communication and coordination among health system actors, poor clinical practices, and gaps in provider training [12]. A study conducted in Kenya identified insufficient training, staff and drug shortages, long queues, limited space and lack of patient-friendly services as some of the facility challenges in implementing lifelong ART among women on lifelong ART [16]. Although certain strategies have been implemented to address some of these challenges, a number of challenges such as antiretroviral drug (ARV) stock-outs, long distances to health facilities, low male partner involvement, and non-retention in care persist [17–19]. Relatedly, a review conducted to assess progress, gaps and research needs towards achieving UNAIDS targets for pregnant and postpartum women in SSA identified a number of research priorities which included; barriers to ART uptake, retention in care, and sustained ART adherence [20].

There is therefore a need for contextual studies to explore the challenges and countermeasures to inform policy and programmatic improvements. To that effect, this study aimed at exploring the health providers' perspectives of challenges along with the corresponding countermeasures to inform implementation of lifelong ART among HIV positive pregnant and breastfeeding women in central Uganda. Countermeasures are strategies that were used to overcome the challenges. Understanding the challenges and countermeasures in implementation of lifelong ART is critical to the success of the PMTCT of HIV program. The findings have potential to strengthen PMTCT programs to attain validation status for, achieve, and sustain e-MTCT of HIV and consequently contribute to the goal of ending AIDS as a public health threat by 2030 [21, 22]. The results are also relevant for similar contexts and in countries that are still experiencing similar challenges in implementing the current policy of test and treat [3, 23–25].

## Methods

### Study design

A qualitative descriptive study was conducted among PMTCT services providers to elicit information on work challenges and countermeasures experienced during implementation of lifelong ART among HIV positive pregnant and breastfeeding women in central Uganda.

### Study sites

The study was conducted in six public health facilities from three largely rural districts. The study sites were: Katikamu Health Centre (HC) III and Luwero HC IV in Luwero district; Mityana General Hospital (GH) in Mityana district; Ssunga HC III, Kyanamukaka HC IV and Masaka Regional Referral Hospital (RRH) in Masaka district. These districts and facilities were selected because they were among the first to implement lifelong ART strategy in Uganda and had fully operational PMTCT programs. The facilities could thus give comprehensive information on challenges and countermeasures. HIV positive pregnant, in labour, or post-delivery women accessed PMTCT services at the antenatal care (ANC), labour wards and postnatal care (PNC) clinics respectively.

## Study population

Fifty-four participants took part in the study as key informants. They were grouped into formal and informal health providers. Formal providers were health workers who had received recognized training with a defined curriculum. They comprised midwives, nurses, counselors, nursing assistants, store assistants (Assistant inventory management officer), dispensers, laboratory assistants, clinical officers, and medical doctors. Informal health providers were health workers who had not received formally recognized training and typically not mandated by any formal institution. Instead, they had some level of training through apprenticeships, seminars, and workshops [26]. Informal providers enrolled in the study were expert clients. Expert clients are people living with HIV who have disclosed their HIV status and are willing to support other HIV clients voluntarily. We included participants who had been involved in lifelong ART services provision to pregnant or breastfeeding women for at least one year. Eligible participants who were too sick or unavailable at the time of the interviews to participate in the study were excluded from the study.

## Selection of study participants

Participants (health providers) were purposively selected based on their roles, workstation, and experience. The aim was to select participants from a broad and diverse sample of providers and clinics to get a comprehensive view on experiences and the organization of services concerning implementing lifelong ART. In total, 54 study participants were included.

This study was part of a larger study on implementation of lifelong ART among HIV positive pregnant and breastfeeding women in Uganda. The objectives of the larger study were to determine the uptake of ART and other PMTCT related services by HIV+ pregnant women and their infants, and to assess retention in care and adherence to ART treatment. Concisely, the larger study enrolled 54 health providers, and 57 HIV positive women for qualitative descriptive inquiry [27, 28]. In addition, we recruited 507 HIV positive pregnant women who were prospectively followed during pregnancy, labour, after delivery and during breastfeeding for a total of 18 months to collect quantitative data [29, 30]. In the current manuscript, we analyse data from the 54 health providers to explore providers' perspectives on the challenges encountered and countermeasures while implementing the PMTCT program.

## Data collection

Data were collected between April and May 2014 using a key informant interview guide. The guide was developed to explore challenges and countermeasures experienced during implementation of lifelong ART among HIV positive pregnant and breastfeeding women. The semi-structured interview guide explored health providers' experiences about challenges in provision of lifelong ART for HIV positive pregnant and breastfeeding women. Besides, countermeasures to deal with these challenges were explored. Table 1 shows the topics, questions and probes that were included in the study tool. The initial tool was pretested by some of the investigators and later discussed by the entire study team. We noted the consensuses and incongruities which we resolved through agreement. This ensured consistency in administration of the semi-structured key informant interview (KII) guide and interpretation of the questions. One-on-one interviews were conducted by four of the lead investigators (A.M., R.W., E.B., and R.N.). Each interview was audio-recorded with consent from the study participant and generally lasted 1.5 hours.

## Data management and analysis

All audio-recorded data were transcribed verbatim; data in Luganda *(local language)* were concomitantly translated and transcribed into English. Final transcripts were stored securely on

**Table 1. Topics, questions and probes that were included in the study tool.**

| Topic | Questions | Probes |
|---|---|---|
| Challenges faced in supporting women on lifelong ART | What challenges do you face in supporting women on lifelong ART? | •Support for retention, adherence.<br>•Drug stock-outs, staffing numbers, and trainings/skills<br>Early infant diagnosis (EID) of HIV |
| challenges faced by pregnant women on Option B+ | In your experience, what challenges do pregnant women on lifelong ART face? | • Loss to follow up, ART adherence, keeping clinic appointments, drug stock-outs.<br>• Stigma, discrimination<br>• Referrals across difference service points such as ANC and ART clinics. |
| Countermeasures to address the challenges | What measures has the health facility/clinic instituted to address the challenges and how effective are these strategies? | • Support for retention, adherence.<br>• Drug stock-outs, staffing numbers and trainings/skills<br>• EID<br>• Loss to follow up, ART adherence, keeping clinic appointments.<br>• Stigma, discrimination<br>• Referrals across difference service points such as ANC and ART. |

password-protected external drives. Data analysis was done using thematic analysis [31] and comprised the following steps: familiarisation with the data by reading through all the data repeatedly to understand the data. A list of ideas from the data was generated, and this was used to develop codes for analysis which A.M., H.B., and R.W. later discussed, unified and organized around challenges and countermeasures in implementation of lifelong ART among pregnant and breastfeeding women. All transcripts were exported to Atlas software (Atlas.ti, Version 7 software, Berlin, Germany) to guide the coding process. Eventually, the codes were sorted into subthemes, categories, and themes. Analysis was undertaken by A.M., H.B., and R.W. regularly evaluated and discussed the entire process of analysis and output for quality control and to ensure that the interpretation was close to the content and supported reflexivity. Illustrative quotations have been presented to enhance the study findings.

## Ethical considerations

Makerere University School of Public Health Higher Degrees, Research and Ethics Committee and the Uganda National Council for Science and Technology approved the study. Permission was also obtained from the study districts and facilities. Participants were assured of anonymity and confidentiality: interviews were conducted in a private environment and transcripts did not bear participant names, nor any other identifiable details. Written informed consent was obtained from each study participant. The informed consent included a section on publication of anonymized responses. Each participant received compensation of 10,000 Uganda Shillings (equivalent to 4USD at the time of the study) for their time.

## Results

### Characteristics of study participants

Table 2 shows the characteristics of health providers who participated in the study. Overall, 54 interviews were conducted with 22 midwives, 8 nursing assistants, 7 expert clients, 4 doctors, 4 store assistants, 3 clinical officers, 3 nurses, one counsellor, one dispenser and one laboratory assistant. Majority of the participants were female (76%, 41/54), while half had received training in provision of lifelong ART to pregnant and breastfeeding women. Participants had worked in HIV care for a median (IQR) of 5 (1–8) years.

**Table 2. Characteristics of study participants (n = 54).**

| Characteristic | Number (%) |
|---|---|
| **District** | |
| Masaka | 34 (63.0) |
| Mityana | 10 (18.5) |
| Luwero | 10 (18.5) |
| **Health Facility** | |
| Masaka RRH | 12 (22.2) |
| Mityana GH | 10 (18.5) |
| Kyanamukaka HC IV | 10 (18.5) |
| Luwero HC IV | 10 (18.5) |
| Katikamu HC III | 7 (13.0) |
| Ssunga HC III | 5 (9.3) |
| **Sex** | |
| Female | 41 (75.9) |
| Male | 13 (24.1) |
| **Cadre** | |
| Midwife | 22 (40.7) |
| Nursing assistant | 8 (14.8) |
| Expert client | 7 (13.0) |
| Doctor | 4 (7.4) |
| Store assistant | 4 (7.4) |
| Clinical officer | 3 (5.5) |
| Nurse | 3 (5.5) |
| Counsellor | 1 (1.9) |
| Dispenser | 1 (1.9) |
| Laboratory assistant | 1 (1.9) |

## Providers' perspectives of challenges and countermeasures

The challenges are arranged under three themes of; inadequacy of HIV service delivery, non-utilization of HIV services, and suboptimal treatment adherence. Under each theme; categories, sub-themes, and countermeasures are presented as shown in Fig 1.

**1. Inadequacy of HIV service delivery and countermeasures.** Two categories were identified under this theme: health worker-related; and logistics, and infrastructure-related.

*a) Health worker-related.* Four sub-themes were identified under this category.

Lack of relevant training

Half of the participants had not received training in the provision of ART for pregnant and breastfeeding women. Additionally, most interviewees reported a complete lack of training or lack of refresher training in lifelong ART provision among their colleagues. Among those directly involved in providing maternal, neonatal, and child health services; most of those who had not been trained were new at the facility. Participants also mentioned that, providers who were not directly involved in providing maternal, neonatal and child health services lacked relevant PMTCT training. These included laboratory, pharmacy and store staff.

*"As a supplies officer [Store assistant], I was left out when trainings on lifelong ART for HIV positive pregnant and breastfeeding women were going on. They brought these drugs and they just told me that these are lifelong ARVs"* Health Provider, Hospital.

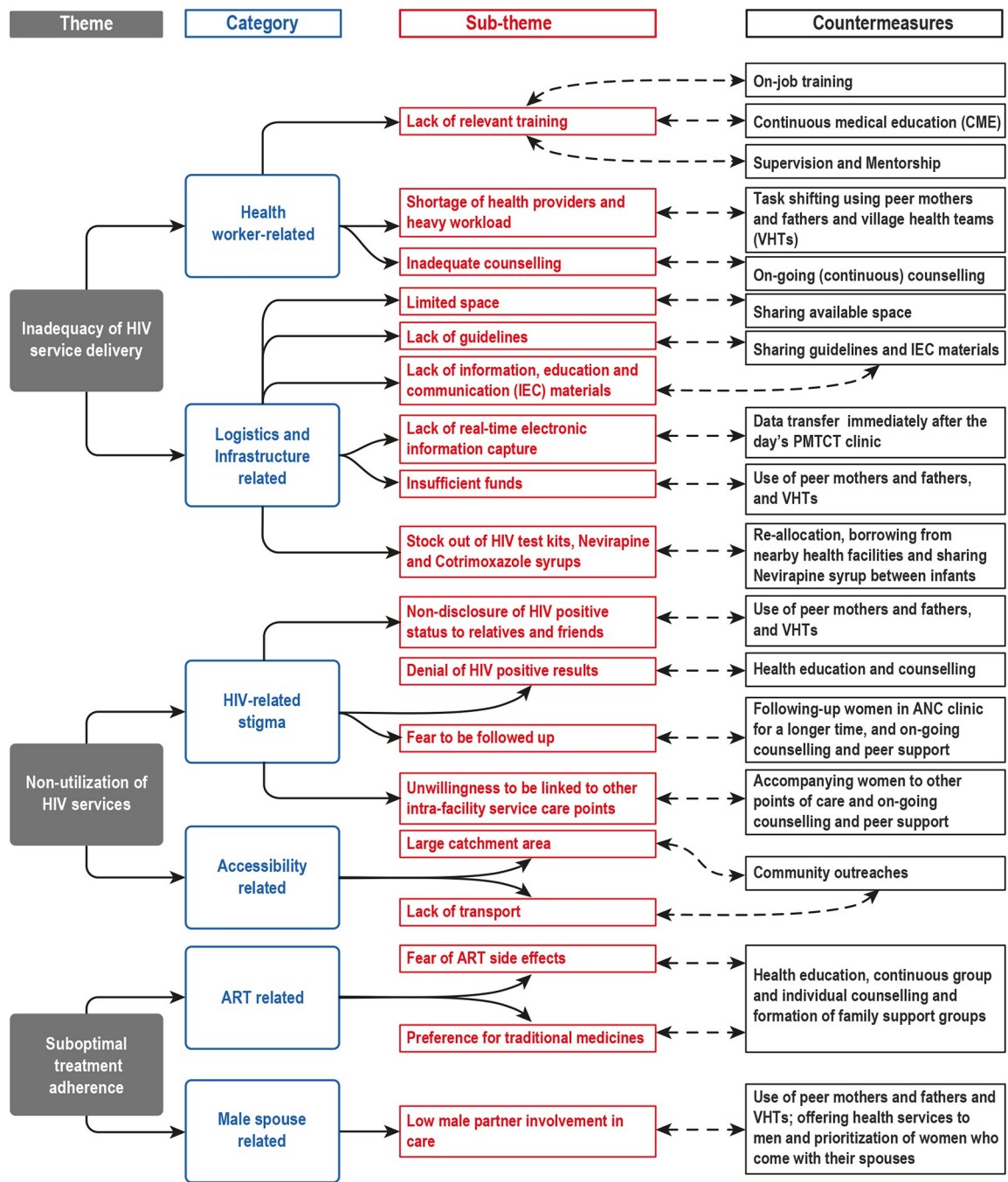

**Fig 1. Illustrates the themes, categories, sub-themes, and countermeasures in implementation of lifelong ART services for PMTCT in three districts of Central Uganda.**

All participants underscored the importance of training more providers to build capacity for sustainable lifelong ART service provision. To address the training gaps, study participants mentioned several strategies that were being used to build provider capacity. These included: on-job training, continuous medical education (CME), and supervision and mentorship by the MOH, district health officials, and implementing partners including Mildmay Uganda and

Protecting Families Against HIV/AIDS. Supervision and mentorship were being done once every three months.

> *"Supervision and mentorship are conducted quarterly [once in three months]. The ideal would have been every month but due to resource constraints, this is not possible"* Health Provider, HCIV

Shortage of health providers and heavy workload

Some participants highlighted that there were low numbers of some providers such as counsellors and clinical officers. In those cases, midwives were overloaded as they were expected to provide counselling, and maternal, neonatal, child and health services to many female clients and their children. The clinical workload was exasperated by the numerous PMTCT paperwork. The ensuant long queues affected staff attitudes who at times became flustered, and exhausted. Eventually, clients would experience delays at clinics.

> *"There are usually many clients [HIV positive individuals] yet we have very few health workers. We may have just three health workers and as one is immunizing the babies, one may be filling various cards and registers while the other is examining the pregnant women and at the same time checking their cards. . . . . . . the queue may be so long and there is no way she can talk to one lady for so long"* Health Provider, Hospital.

Participants noted that health facilities adopted task-shifting using expert clients and village health team members (VHTs), to overcome staff shortages and heavy workload. Task shifting was observed as being very helpful in reducing the heavy workload and patient waiting time.

> *"We have expert clients who are also peer mothers in the ANC clinic. They help to counsel these mothers and to follow them up. They support us a lot in caring for the mothers on lifelong ART"* Health Provider, HC IV.

Inadequate counselling

All participants highlighted the importance of adequate and quality counselling to ensure lifelong ART uptake, adherence and retention in care by the women and their infants. However, many felt that the counselling offered was inadequate due to a lack of counselling training, a shortage of dedicated counsellors, and limited time to counsel women on the range of issues, amidst an overwhelming number of women who required the service.

> *"Counselling is not enough; the workload is too much and the counsellors are very few yet, we have bigger clinics, the PMTCT and the ART clinics. I think we have about two counsellors and the other health workers provide the counselling but it is not really conducted satisfactorily. If we don't give the mothers continuous information, they will become reluctant."* Health Provider, Hospital.

Inadequate counselling was addressed through offering continuous (ongoing) counselling to women at every facility visit with emphasis on issues at hand.

> *"We offer ongoing counselling to HIV positive pregnant and breastfeeding mothers on lifelong ART every time they come back. We ask them the challenges they have met when they are at home taking the lifelong ART and then offer appropriate support."* Health Provider, HC IV.

*b) Logistics, and infrastructure-related.* Under this category, six sub-themes were identified.
Limited space

Limited space was a prominent challenge cited by most participants. This was mostly attributed to the increased number of women seeking services, yet different services that had been integrated into the clinic were still offered in the same space. This resulted in congestion, which compromised privacy and confidentiality. Mothers were reportedly uncomfortable receiving services in such an environment.

*"The place where they get the information is open and small, near a walk-way and there are so many activities taking place, it is extremely small. So, there are so many things that might divert their attention [when counselling is going on], we don't have any privacy"* Health Provider, Hospital.

Limited space, especially for counselling was difficult to address because it was affecting all other departments and not only limited to maternal, neonatal, and child health clinics. Providers said that they continued to work in the available space but wished that government and implementing partners would set up new buildings or temporary structures like tents.
Lack of guidelines

Most participants noted that copies of PMTCT guidelines were limited. In some facilities, participants reported never seeing any lifelong ART guidelines for pregnant and breastfeeding women. Where guidelines existed, health providers kept them in their consultation rooms and shared them when need arose.
Lack of information, education and communication (IEC) materials

PMTCT IEC materials were also lacking except for a few facilities that had such information on the walls.

*"Yeah, there are some posters, but they are extremely few. I have seen one, but recently, they repainted the walls and I don't know if they have put them up again"* Health Provider, Hospital.

Additionally, participants noted that facilities lacked fliers and information packs that could be provided to mothers. A few participants felt these materials could be taken home by the women to assist their understanding and to enable them share with family members especially their spouses.

*"It would be good to give HIV positive mothers materials such as leaflets with lifelong ART information because it reminds the woman to take the ARVs. Additionally, if put on the table—and my husband doesn't know that am HIV positive; when my he comes and reads it, he will get the information and will realize that there is need to test for HIV"* Health Provider, HC IV.

Lack of real-time electronic data capture system

In all the health facilities, participants highlighted that lack of real-time electronic data capture system was a challenge. They reported that they recorded patients' data in PMTCT registers and on cards which were later given to staff in the records' section for entry. Participants emphasized that time from data capture to electronic entry depended on the volume of patients and number of record assistants available. Once the data had been electronically entered it was used to track mothers and later their babies as well.

*"We record the mother's information on paper. Thereafter, we send it to the in-charge who verifies the information captured and then takes it to the records assistants for entry into the computer"*. Health Provider, HC III

Insufficient funds

Many participants indicated that limited funding hindered making of phone calls, home visits and community out reaches to follow up women and their infants who missed clinic appointments.

*Follow-up and tracking of women and their exposed infants require funds for transport. We at times use bicycles but they are few and some places are far away. I wish the Ministry of Health comes to provide more funds, bicycles, and motorcycles"*. Health Provider Hospital

Participants said that in addition to use of VHTs and expert clients, other strategies like health facility providers using facility and district vehicles, follow-up and track women despite the limited funding.

*"We give the HIV positive women addresses of the VHTs to go and follow-up on them. The VHTs go to check whether the mother and baby are still there and why they are not coming for their clinic visits or taking the ART"* Health Provider, Hospital

Stock-outs of HIV test kits, Cotrimoxazole and Nevirapine syrups
Most participants observed that health facilities occasionally faced shortage of HIV testing kits, Nevirapine and Cotrimoxazole syrups. However, it was acknowledged that at one point the shortage of testing kits was a country-wide challenge. Participants admitted that sometimes stock-outs were due to poor forecasting which was rectified by devising a mechanism for enhanced forecasting.

*"We have been experiencing stock-outs for; testing kits and the syrup—baby syrup [meaning Nevirapine]"*. Health Provider, HC IV

Internal re-allocation of supplies and borrowing from nearby facilities were used to address this, and at other times, dividing of drugs was done.

*"We would request Nevirapine syrup from the lower-level health units which are not very busy. . . . . . . . . It reached a moment where we had to share the Nevirapine syrup among two babies"*. Health Provider, Hospital

In cases where providers were completely constrained, they would inform the implementing partners who would then bring the logistics.
**2. Non-utilization of HIV services and countermeasures.** Two categories are presented: HIV-related stigma and accessibility related.
*a) HIV-related stigma*. Four sub-themes are presented under this category.
Non-disclosure of HIV positive status
All study participants expressed that non-disclosure of HIV positive status was a significant and common challenge. They highlighted that many HIV-positive women feared disclosing their HIV status and the women perceived it that they would be stigmatized. It was noted that some women didn't disclose their HIV positive status to their partners since they weren't sure

of their spouses' reaction to the HIV+ result. Of most concern to the women, was fear of negative impact on relationships.

> *"The man [spouse] will chase me away or run away from me, yet this is still a new marriage!"* Health Provider, Hospital.

> *"She will then tell you, 'ah'! Musawo [health worker] leave me alone, what if I tell him [spouse] and he tells me that am the one who brought the virus!" Another one will say, "what if he [spouse] chases me away with my child.…….. just leave me alone". Actually, most of them do not tell them [spouses] about their HIV positive status and being on ART"* Health Provider, Hospital.

The women ended up hiding their ARVs, missing appointments and not adhering to ART.

> *"The first challenge is non-disclosure of HIV positive status! Mothers were initiated on lifelong ART during ANC and post-delivery, but even after all the counselling is done, some mothers refuse to disclose the positive HIV status to their partners. This mother's ART adherence may be affected because she may not be able to take the drugs when the husband is around. Secondly, she may fear to keep the drugs with her in the house"* Health Provider, HC IV.

Denial of HIV positive results
During interviews, many participants reported that some women did not believe they were HIV-positive. Such women reportedly repeated the test at other facilities which delayed ART initiation.

> *"One woman said that she has been delivering all her children normally [meaning children who were HIV negative] and now we are telling her that she is HIV-positive? How comes? In other words, she doubted her HIV status"* Health Provider HC III.

Fear to be followed up
Participants noted that some women did not welcome being visited or followed up in the communities where they lived due to HIV-related stigma with some providing incorrect addresses and/or telephone numbers. Sometimes women switched off their phones. This affected follow-up and linkage to HIV community-based services.

> *"Yes, we try to call them, but it is hard to find them. When they get to know your telephone contact, they stop picking your calls! Sometimes, when you call them [women], they will tell you that they are busy. Others may tell you that they shifted to another place for the same services"* Health Provider HC IV

Participants said that the use of ongoing counselling, expert clients and VHT support were the strategies in place to overcome non-disclosure, and denial of HIV positive results, and fear to be followed-up.
Unwillingness to be linked to other intra-facility service care points
There are established mechanisms to link women and their infants to different intra-facility service care points, however, several challenges were cited by study participants. For example, some women were unwilling to be linked for intra-facility HIV services mainly because of HIV–related stigma. Many participants reported that some women feared being seen in general HIV clinics by relatives, friends and or neighbours.

*"After giving birth, the mothers are supposed to attend the EID and general HIV clinics from the other side and there are many other patients seated together there. So, some fear and are unwilling to be linked to those other clinics. They prefer to stay at the PNC clinic where they are few and receive the clinic services quickly"*. Health Provider, Hospital.

To overcome this challenge providers cited various strategies. The time of follow-up within the PNC was extended at several facilities to address this lack of willingness to transfer to general HIV chronic care clinics. In four of the six facilities, women were staying in the PNC beyond the recommended six weeks after giving birth.

*"Actually, we continue keeping them within the PNC clinic until the women are ready for referral/linkage to the ART clinic successfully before they deliver or after they have delivered"* Health Provider, HC IV.

As another strategy, women were accompanied by either a midwife, a counsellor, an expert client, or a VHT member from one service point to another, to ensure successful linkage. For example, women would be accompanied from the ANC clinic to the infant immunization clinics or to the EID testing care point and general HIV clinic.

*"Mothers who come here for their first time don't know us or the different health service care points. I move with the mother showing her around and what is done until we reach the EID care point. There is a person responsible for EID, and then I introduce that medical worker to the mother"* Health Provider, HC IV.

*b) Accessibility related.* Under this category, two sub-themes were identified. These were:
Large catchment areas
Some participants reported that they serve women who came from distant places to the study facilities. Facilities especially in rural areas that were close to the women didn't offer lifelong ART services.
Lack of transport
Most HIV-positive pregnant women could not afford transport costs which impacted their ability to attend appointments.
*Some women come from very far*! *They stay like 20 miles away from here. Such women have a challenge of transport coming here every month."* Health Provider HC IV.
To address accessibility-related challenges study participants noted that they used outreaches during which women on lifelong ART would be clinically evaluated and appropriately treated. Additionally, community members were engaged in health education and, HIV counselling and testing (HCT) which was envisaged to reduce community-related-HIV stigma. Besides, expert clients and VHTs would support the women to cope with the lifelong ART.

*"We have three outreaches in a month. We have a number of landing sites where we conduct outreaches. We have our "Kizindalo" [big loud speaker] that we use to call and mobilize them. When they come, we counsel and test them. Those who test HIV positive, are counselled and enrolled into care, while those who test negative are only counselled. We use the occasion to reduce stigma tendencies through health education"* Health Provider, HC IV.

However, it was noted that outreaches were expensive and time-consuming which calls for more resources and consequently result in sustainability challenges.

Furthermore, some participants reported that some female clients were given additional ARV supplies to reduce visits to clinics for collection of medication. This was anticipated to reduce on the transport costs.

*"Some women who come from far are given ARVs for two months instead of one to reduce on the frequency of their clinic appointments. In addition, we have been provided with some fuel to take the ARVs and do HIV testing to the community through outreaches"*. Health Provider, Hospital.

**3. Suboptimal treatment adherence and countermeasures.**   The categories under this theme were ART-related, and male spouse related.

*a) ART-related*. Two sub-themes are presented.

Fear of ART side effects

Most participants said that many women feared ART side effects, and some of which were reported to have been grave.

*"She [HIV positive mother] refused ARVs claiming that she will get a skin rash or even die. At least, she would take Septrin [Cotrimoxazole] but not ART]. She [HIV positive mother] tells you that some of her relatives suffered from HIV and when they started taking ARVs, they got severe side effects and some actually passed away"* Health Provider, HC III.

Preference for traditional medicines

Several study participants mentioned that some women would stop using ART and opt to use local herbs with a misconception that they would treat HIV as illustrated by the quote below;

*"Some women stop taking their HIV drugs and resort 'budomola' meaning the local herbs used to treat HIV. Such mothers continue taking those herbs. But for us [health providers] here, we don't support them to use those herbs because when they will give birth to HIV positive babies in the long run"*. Health Provider, HC III

Participants emphasized that fear of ART side effects and preference for traditional medicines were addressed using similar strategies. These included health education, continuous group and individual counselling, and formation of family support groups (FSGs).

Participants noted that FSGs had helped significantly in addressing these challenges through peer support.

*"Family support groups are helpful; when one is HIV positive, alone and gets a problem, it disturbs a lot but when one is in a group and another person mentions the very problem you are encountering, then you will realize that you are not alone. As a result, your mind is settled"* Health Provider, HC III.

*b) Male spouse related*. Low male partner involvement in care was identified as the sub-theme under this category.

Low male partner involvement in care

Low male partner involvement in care was a common finding. Participants noted that some women would even bring different men other than their partners whenever they are told to bring their spouses. This was attributed to fear for consequences of HIV positive status disclosure.

*"We tell the women to come with their husbands for HIV counselling and testing, and treatment but majority don't want. The mothers tell you that the men are busy while others are scared to come to health facilities".* Health Provider, HC III.

Some men were reported to be unsupportive during ANC, delivery and PNC. Partner support related challenges intensified after delivery especially when the woman did not disclose her HIV status.

*"We have very few husbands that accompany pregnant women and my fear is that when one doesn't have a treatment partner, there is nobody to share with the challenges they are facing. If the man is not involved, what will happen if that woman depends on the man for financial support? At first, it was the pregnancy driving the man to give money to the pregnant mother to come for ANC, now that the baby has been delivered, what will be the reason for the husband to give transport to this mother to come for ART?"* Health Provider, Hospital.

There were various interventions across facilities to overcome low male involvement. In most facilities, women who came with their husbands were prioritized for health services. However, in other instances, men would actively be involved in the care that the wives received.

*"Apart from health education that we provide, the men are there with us, when we are taking body weight and blood pressure; we also take theirs (the men). Inside the examination room, we also ask the husband if he is having any sickness and we give them the treatment. Moreover, after I examine the mother, I ask the spouse, [have you ever felt your baby's heart beating from the uterus?] Then the person says no. Then I ask, "Do you wish to feel it beating?" When he agrees, I just put a foetal scope there, call him and give him instructions. When the husband hears the baby's heart beating, you see that he is really excited"* Health Provider, HC IV.

Participants reported providing health services to men such as health education, screening for, and treating common illnesses motivated the men to accompany their wives to the health facilities. In other facilities, study participants said that they didn't prioritize those who came as couples but thanked and recognized men who had accompanied their wives. Other strategies to improve male involvement encompassed community outreach, continuous counselling, health education, use of FSGs and peer fathers/male champions.

*"We realized that some of the strategies that we are using motivate men to come with their spouses for lifelong ART services. The strategies include giving health education on topics like HIV, care for pregnant and breastfeeding women, medical male circumcision, sexually transmitted infections (STIs) and non-communicable diseases. Other approaches that we use are offering men health services such as screening for STIs, non-communicable diseases like hypertension and diabetes mellitus".* Health Provider, Hospital.

One participant said that they were exploring the possibility of encouraging men to be involved in care for their spouses. He suggested that they were going to target men when they come to visit their wives in the health facility which is common in the first few hours after delivery. This would be an opportunity to encourage male involvement since most men come to check on their wives and the new-born infants.

*"Normally when the woman delivers, that is when her husband comes to see or pick her and the new-born. So, we are making an attempt to see whether we can use that as a time to*

*counsel men as well. It is an opportunity that we have, to talk to the men"* Health Provider, HC IV.

## Discussion

This study explored challenges and countermeasures in the implementation of lifelong ART for pregnant and breastfeeding women from the health providers' perspective in three districts in Central Uganda. Major challenges were: a lack of in-service and/or refresher training on lifelong ART, shortage of health providers, inadequate counselling, lack of real-time electronic data capture system, and stock-outs of HIV test kits, Nevirapine syrup and Cotrimoxazole syrups which might result in inadequate lifelong ART service delivery. Other major challenges such as non-disclosure of HIV positive status, fear to be followed up, large catchment area, and lack of transport suggest non-utilization of lifelong ART services. While challenges like fear of ART side effects, and low partner involvement in care could lead to suboptimal treatment adherence lifelong ART among pregnant and postpartum women.

Several strategies were being used and proposed to address the identified challenges. These included on-job training, continuous medical education, task shifting, ongoing counselling, FSGs, re-allocation of HIV commodities, use of peers, and conducting community outreaches.

The Uganda MOH working with U.S. President's Emergency Plan For AIDS Relief (PEP-FAR) and implementing partners have addressed some of the challenges such as training inadequacies, stock-outs, retention in care challenges, and staff shortages to ensure that Uganda remains on track to achieve both the UNAIDS and national 95-95-95 targets. Strategies that were employed to address the gaps include increasing funding for ARV procurement and stock monitoring, supply chain reform improvement, human resource performance management, improving referral and linkage structures, use of short message services, phone calls, and home visits to track the lost women and HIV exposed infants, bring back mother and baby campaign, and supporting outreaches [19, 32, 33]. Recent reports indicate that several of these challenges still exist including inadequate number of health workers, increased workload, lack of training, insufficient supplies and commodities, suboptimal ART adherence, non-disclosure of positive HIV status, low retention in care rates and low male partner involvement in care [17, 34–36]. Likewise, we earlier reported challenges around adherence and non-disclosure of HIV positive status from the clients' perspective [27, 30]. Therefore, this study adds to existing literature on challenges in implementation of lifelong ART among pregnant and breastfeeding women from the heath providers' perspective. The study also describes approaches that were used to address the challenges. Addressing these persistent challenges will fast track the achievement and sustainability of e-MTCT outcomes towards ending AIDS as a public health threat by 2030 [21, 22].

### Lack of relevant training and inadequate counselling

Quality of e-MTCT services continues to experience some flaws which should be addressed [35]. Challenges that result in inadequate HIV service delivery might be contributors to low quality e-MTCT services. Many health providers had not received training on PMTCT lifelong ART service provision. This was mostly among providers who were new in the facilities and those who were not directly involved in offering lifelong ART services. On the other hand, some of the previously trained providers lacked refresher training. Congruent with our findings, a study conducted in India, found that some health providers lacked training on PMTCT services [18]. Other studies including a systematic review and another study done among health care providers at four health facilities in western Kenya revealed that the PMTCT

training received by health providers was insufficient [12, 16, 37]. Furthermore, despite the adoption and implementation of the innovative differentiated service delivery (DSD) models for HIV services in Uganda in 2016, a study found that health workers in 600 out of 1,800 health facilities providing ART had not yet been trained in DSD delivery [36]. This suggests persistent gaps in HIV service delivery training in Uganda. Our study finding calls for e-MTCT training of more health providers and refresher trainings for those who were already trained. Innovative trainings such as web based, virtual and modular would be more appropriate and cost-effective since they reduce movements of health providers from their work stations. Additionally, the web based and virtual designs could be appropriate in the current context of COVID-19 pandemic.

Moreover, the trainings could be designed to address gaps in counselling that our study identified as one of the major challenges. Although adequate counselling was earlier reported to be the reason for the women to start swallowing ART on the same day it was prescribed [29], this might have targeted the newly diagnosed HIV pregnant women. The counselling inadequacies might be partially attributed to lack of or insufficient training on e-MTCT services, limited time for counselling, overwhelming numbers of individuals who require HIV services, and a lack of trained counsellors. Adequate counselling is essential especially in preparing pregnant and breastfeeding women for lifelong ART [38]. It prepares the women for the lifelong journey of ART. In contrast, inadequate counselling results in low utilization of PMTCT services [39] which would deter the achievement of e-MTCT of HIV. Most challenges identified such as the denial of HIV+ results, fear of ART side effects, fear to be followed up and non-disclosure HIV status can be solved using good counselling skills.

### Shortage of health providers

Shortage of health providers was another challenge identified by this study. Staffing inadequacy in Uganda is not only a PMTCT challenge but rather cuts across various programs [40]. Notably, our study identified counsellors and clinical officers as some of the health providers who were inadequately staffed. Unfortunately, this shortage in the PMTCT program is worsened by the extensive patient and documentation load associated with HIV service provision [41]. Similar to our study, a validation study on the path to e-MTCT of HIV and Syphilis in Uganda (2010–2018) showed that at facility and district level, 50% (64) and 54% (32) respectively, had inadequate staff in both category and numbers offering PMTCT services [17]. Correspondingly, a study in Nigeria revealed a huge lack of doctors, nurses and midwives to available to deliver PMTCT services [23]. Task shifting using non-formal health providers such as expert clients (peer/mentor mothers and fathers), community linkage facilitators and VHTs has been adopted to address this challenge. However, the number of this cadre of health workers remains low [42]. Non-formal providers might have more time with women on lifelong ART because some of them live within the same communities. Expert clients have lived HIV experience, placing them in a better position to handle HIV-related stigma challenges. Nevertheless, the use of a mix of formal health workers, expert clients and other informal volunteer workers calls for more innovation to ensure delivery of quality services and sustainability [43–45]. The MOH and implementing partners should ensure adequate training, supervision and mentorship of expert clients, linkage facilitators and VHTs to enable them work effectively.

### Lack of real-time electronic information capture, guidelines, and IEC materials

Absence of real-time electronic data capture system and limited supply of PMTCT guidelines and communication materials was noted as a shortfall. This finding is similar to results from a

study in Malawi where health workers reported lack of information technology infrastructure which resulted in use of paper-based methods of data capture and storage [46]. Paper-based data has limitations such as storage, unreadable, double work, time consuming, and errors during entry into the electronic medical record system, and difficulties in analyses [46, 47]. In India, it was found that real-time electronic data capture systems improve health provider work style, data capture, linkage across different clinics and overall satisfaction with PMTCT work [48, 49]. MOH in collaboration with the donors should strengthen scale up of electronic medical records to enable real-time data capture, easy and timely analysis to inform decision making. Additionally, more copies of PMTCT guidelines should be distributed to health facilities to enhance standardized practice. Participants felt that PMTCT communication materials would be useful in disseminating information to individuals, families and communities especially men who are reluctant to visit health facilities. Client IEC materials have the potential to reduce HIV related stigma and hence increase utilization of PMTCT of HIV services [50]. Appropriate, context specific and easy to understand client PMTCT IEC materials should be availed to health facilities and communities.

## Stock-outs of HIV test kits, Nevirapine and Cotrimoxazole syrups

Our findings show occasional stock-outs of HIV test kits, Nevirapine and Cotrimoxazole syrups. This is consistent with findings from a study in India, where study participants reported shortages of HIV testing kits, and Nevirapine for infants [18]. However, a study from health facility surveys in six SSA countries had varied results. Stock-outs of HIV test kits and ARVs were common in Tanzanian health facilities [51]. Stock-outs might suggest poor forecasting, management, and monitoring of these essential HIV commodities. A lack of HIV testing kits, Nevirapine and Cotrimoxazole syrups is unacceptable. HIV testing is the entry point for primary and secondary prevention as well as care and treatment, while the ARVs supply chain is a major component of a systematic program for the prevention and treatment of HIV/AIDS [52–54]. Health facilities with stock-outs would make arrangements to get these from facilities that had enough stock. This is in line with the MOH redistribution strategy [7]. Although the Uganda MOH with support from partners is closely undertaking ARV stock monitoring and management using the Web-Based ARV Ordering and Reporting System [19], these findings underscore the need for all e-MTCT of HIV stakeholders to work together to avert the recurrent stock-outs of HIV commodities. This could be done by strengthening the supply chain management capacity at national, regional, district and health facility levels which would ensure accurate and timely ordering and provision of appropriate real-time stock status data at health facilities.

## Unwillingness to be referred

Health providers noted that some women refused to be referred to other HIV intra-facility service care points. The women preferred to remain in the antenatal clinic even after giving birth which was attributed to HIV- related stigma. This finding is similar to results from a study that was done in Nigeria. Participants reported that HIV positive women feared to get services from designated places in the health facilities [55]. Refusal to receive services from other HIV care points is likely to result in loss-to-follow up along the PMTCT cascade. Facilities used various approaches to ensure that women are not lost during the linkage process. Strategies used included accompanying women from one service point to another, following up referrals that were given to women and only referring women when they were ready. This was to ensure that women reach the point of referral. Howbeit, accompanying women could be a challenge and not feasible where health providers face staff shortages and heavy workload.

## Large catchment area and lack of transport

Despite the target by the Uganda MOH that 85% of population to reside within a distance of 5km from a health facility, distance to health facilities remains a challenge [56, 57]. Our study highlights the challenge of large catchment areas and lack of transport. These findings concur with those from Ethiopia and South Africa where restricted physical access, and transportation to PMTCT sites were barriers to utilization of PMTCT services [58, 59]. Expert clients, linkage facilitators, VHTs and community outreaches were being employed to address this challenge. Nonetheless, these strategies are costly and unsustainable. Given that several women and other clients on test and treat attend HC IIIs and IIs for both ANC and immunization yet these facilities often lack comprehensive HIV services, the MOH ought to strengthen HIV services at HC IIIs, accredit large volume HC IIs to provide comprehensive HIV services.

## Low male partner involvement in care

Men are the main decision-makers in a home especially in Africa [60, 61]. They thus influence women's access to maternal, neonatal and child health services, and thus impacting their health outcomes [61]. Male involvement in implementation of the lifelong ART strategy affects uptake, adherence and retention in care [62–64]. Low male partner involvement in care for lifelong ART among pregnant and breastfeeding women stood out in our study findings as a challenge. Consistent with our study, findings from studies conducted elsewhere show low male partner involvement in care [65–67]. For example, in Mwanza region–Tanzania the study showed that only 24.7% of mothers indicated that their male partners were involved in PMTCT, whereas only 20.9% of men from a study in Ethiopia had a high involvement index in PMTCT services [66, 67].

Facilities in our study were engaging some strategies to improve male involvement in lifelong ART services but providers reported that the impact was limited possibly because they were still in infancy stages. Strategies include community sensitization of men about benefits of ANC and lifelong ART, having male friendly services for men who accompany their spouses and offering quality services. Strategies such as use of mentor fathers/champions, health education, community sensitizations and screening of non-communicable diseases for men might have a great potential to improve male involvement in lifelong ART services. Regrettably, male partner involvement has remained a gap in the PMTCT of HIV program [17]. The countermeasures that our study identified to address low male involvement could be implemented in other African country settings so as to accommodate male partners in attending, ANC, delivery, and PNC services with their female partners. Albeit, more innovations such as age and population specific dialogues are needed to overcome low male partner involvement in e-MTCT of HIV.

## Strengths and limitations

A broad range of providers implementing the PMTCT program were interviewed which enabled us to explore challenges and countermeasures within the facilities. The data was collected in 2014 and some of the findings may not be applicable in the current context. In addition, the PMTCT program has gone through a number of transitions in the last decade. For example, the last transition is the change to Dolutegravir ART-based regimen [11]. Viral load monitoring has become the standard of care compared to CD4 count. These changes may have implications on our results. However, several of the findings remain relevant and can be transferred to other facilities and settings with Uganda and beyond. Use of key informant interviews might have resulted into social desirability bias but this was minimized by use of experienced data collectors, establishing of rapport, use of probes, and conducting regular research team meetings.

## Conclusions and recommendations

The study identified persistent challenges that were negatively impacting HIV service delivery and use, and subsequently treatment adherence. A number of countermeasures were being used to address these challenges. We recommend that; providers including community support groups offering lifelong ART services to pregnant and postpartum women should receive regular training, supervision, and mentorship. Training on lifelong ART services should start during the pre-service period for all clinical students. Adequate personalized pre-ART and continuous counselling should be offered to HIV-positive pregnant and breastfeeding women. Facilities should be supported to ensure that they; are well stocked with lifelong ART and health supplies, and acquire real-time electronic data cap capture system. Ministry of health should scale-up the strategy of task shifting using peers, VHTs, and male champions, and consider accreditation of the large volume HC IIs and those in distant rural areas to provide e-MTCT services. Research to assess the effectiveness of the key countermeasures is recommended so as to inform adoption and scale up.

## Acknowledgments

The authors wish to thank all study participants without whom this study wouldn't have been successful. Special thanks also go to the respective heads of facility for Masaka RRH, Mityana General Hospital, Luwero HC IV, Kyanamukaka HC IV, Ssunga and Katikamu HC IIIs for their continued support rendered to us throughout the data collection process. We also acknowledge the tireless efforts of all study interviewers in collecting data at their respective study sites.

## Author Contributions

**Conceptualization:** Aggrey David Mukose, Fredrick Makumbi, Esther Buregyeya, Joshua Musinguzi, Rhoda K. Wanyenze.

**Formal analysis:** Aggrey David Mukose, Hilde Bastiaens, Rhoda K. Wanyenze.

**Funding acquisition:** Joshua Musinguzi, Rhoda K. Wanyenze.

**Investigation:** Aggrey David Mukose, Fredrick Makumbi, Esther Buregyeya, Rhoda K. Wanyenze.

**Methodology:** Aggrey David Mukose, Esther Buregyeya, Rhoda K. Wanyenze.

**Project administration:** Rose Naigino.

**Supervision:** Hilde Bastiaens, Jean-Pierre Van Geertruyden, Rhoda K. Wanyenze.

**Visualization:** Aggrey David Mukose, Hilde Bastiaens.

**Writing – original draft:** Aggrey David Mukose.

**Writing – review & editing:** Aggrey David Mukose, Hilde Bastiaens, Fredrick Makumbi, Esther Buregyeya, Rose Naigino, Joshua Musinguzi, Jean-Pierre Van Geertruyden, Rhoda K. Wanyenze.

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
