## [Decision Letter · Decision Letter 0]

25 Aug 2022

PONE-D-22-14861Challenges and commonly used countermeasures in the implementation of lifelong antiretroviral therapy for PMTCT in Central Uganda: Health providers’ perspectivePLOS ONE

Dear Dr. Mukose,

Thank you for submitting your manuscript to PLOS ONE. After careful consideration, we feel that it has merit but does not fully meet PLOS ONE’s publication criteria as it currently stands. Therefore, we invite you to submit a revised version of the manuscript that addresses the points raised during the review process. The comments can be found at the bottom of this letter.

We look forward to receiving your revised manuscript.

Kind regards,

Desmond Kuupiel, PhD

Academic Editor

PLOS ONE

Journal Requirements:

“This study was funded by the Global Fund through Ministry of Health - Uganda; Grant Number: 683 UGD-708-G07-H. The contents of this article are solely the responsibility of the authors and do 684 not necessarily represent the official views of Global Fund, Ministry of Health or Makerere 685 University School of Public Health.”

 “This study was funded by the Global

Fund through the Ministry of Health-Uganda [Grant

Number: UGD-708-G07-H]. The funders had no role in study design, data collection and analysis, decision to publish, or preparation of the manuscript.”

Please respond to all the reviewers' comments.

Reviewers' comments:

Reviewer's Responses to Questions

**Comments to the Author**

1. Is the manuscript technically sound, and do the data support the conclusions?

Reviewer #1: Yes

Reviewer #2: Yes

2. Has the statistical analysis been performed appropriately and rigorously? 

Reviewer #1: N/A

Reviewer #2: N/A

3. Have the authors made all data underlying the findings in their manuscript fully available?

Reviewer #1: Yes

Reviewer #2: No

4. Is the manuscript presented in an intelligible fashion and written in standard English?

Reviewer #1: Yes

Reviewer #2: Yes

5. Review Comments to the Author

Reviewer #1: The manuscript describes the challenges and countermeasures in the implementation of lifelong ART In central Uganda. The manuscript is very well written and easy to follow and understand. There are a couple of issues when addressed will further enrich the publication

1. Introduction: It will be useful if information on transmission rates of MTCT in Uganda is included in the overview. This will enable readers put the results in perspective

2. 54 providers from 6 facilities and 3 districts: kindly show in a table the number and categories of providers from each health facility

3. Even though, the authors acknowledge that data from 2014 may be an issue but think that the results may still be relevant. I am particularly concerned because MTCT has gone through a number transitions in the last decade. The last transition is the change to Dolutegravir with the attendant issues family planning particularly in the initial phase. In addition, viral load monitoring has become the standard of care compared to CD4 count. The implication of the results must be situated within the current context . This was not done

Reviewer #2: Thank you for this interesting manuscript. Please do a final editorial review after addressing the comments.

Abstract: Methods:

The authors note that there were 54 key informant interviews with providers from 6 facilities – in the manuscript, it seems that the key informants extend beyond providers, including for example expert clients (which is great for data quality) – please make this clear in abstract. Or is the term “provider” being used to include all these categories of informants?

Introduction:

Historical background to ART initiation in Uganda well documented, and rationale for manuscript well presented.

Methods:

Study sites: Do ethics requirements allow for facilities to be named? Double check study requirements regarding identifiers and protecting confidentiality of providers. Section on ethical consideration says they were assured of confidentiality.

Study population: See comment in abstract – It is a bit confusing who the participants in this study were – the authors say 54 key informant participants were included – and then lists them (Lines 129-131) – then later, Line 137, they say 54 health providers and 57 HIV positive women were enrolled. Are the initial 54 key informant participants the same as the 54 health providers? Please make this description clearer? In the results section, it is confirmed that the 54 key informants are as they are first listed in this section.

Thanks for clarifying what expert clients are. What are store assistants?

Line 133: Sentence starting “This study was part of…” – perhaps this should be the start of a new paragraph? Or clearly distinguish what is part of the study reported on in this manuscript, and what is the bigger study?

Lines 138-140: Once the details of the participants (all providers, or other key informants too) are clarified, make sure that this sentence outlining the aim of the manuscript is in line with this (provider’s perspective only?).

Data management and analysis (and elsewhere): Where author’s initials are used, standardise if they are written with periods/full stops or not (AM vs A.M).

Results:

Figure 1 quality is not very good – quite hard to read.

Countermeasures – are these only described when reported on? It would be helpful to highlight this, as under most challenges they are described, and others there is no mention of how providers deal with the challenges.

Descriptions in results – in some places these can be consolidated and summarised, as the illustrative quote gives the details – to double check that the summary point is not repetitive of what is reported in the quote (e.g. Lines 409-413: the countermeasure description could possibly be summarised as follows: “Participants reported that some clients were given additional ARV supplies to reduce visits to clinics for collection of medication” followed by the quote) – to check all.

Inadequacy of HIV service delivery and countermeasures:

Health worker related: Lack of relevant training:

Line 207: This quote is from a supplies officer, since the different categories of staff were interviewed, it may be helpful to specify in the quote identifier that it was a supplies officer at a hospital, not a health provider?

Are there any quotes/data to support the feedback of supervision and mentorship which were being done every 3 months? The limited training seems to indicate that not many people are exposed to this?

Shortage of health providers:

Line 219: Women – replace with female clients?

Line 220: Elaborate – how do long queues affect staff attitudes? Make them tired, flustered – this is linked closely to heavy workload. If too few staff, workload is more? Describe/distinguish the overlap between these points.

Logistics and infrastructure related: Lack of guidelines

It seems like the guidelines do exist, it is that there are a lack of hardcopies of these guidelines? This also links to training on guidelines – if people are aware of the guidelines, they only need to refer to hardcopies for points of clarification?

Insufficient funds:

What should the source of these funds be? Is it budget allowance from government?

Line 303-305: Who funds these activities if there is limited funding? NGOs? Individual healthcare providers?

Non-utilisation of HIV services and countermeasures:

HIV related stigma: Non-disclosure of HIV positive status

Line 333: perhaps reword – “fear of marriage breakages” to “fear of negative impact on relationships” – this extends to multiple partnerships scenarios, and the example given is about a marriage break-up.

Fear to be followed-up: Is this follow-up referring to treatment adherence follow-up, linkage to care, reminders to come for clinic visits, or what kind of follow up? It would be helpful to know what is being referred to here?

Unwillingness to be linked to other intra-facility service care points:

Line 367: What is EID? Make sure all acronyms in full first time.

Suboptimal treatment adherence and countermeasures:

ART related: Preference for herbs:

Should this section “preference for herbs” be reclassified as “preference for traditional medicines”? Is the use of herbs about using traditional medicines?

Male spouse related: Fear of spousal reaction:

This section seems to be very closely related to the section on stigma and non-disclosure – which was because of fear of spousal reaction, resulting in non-adherence? How were these themes differentiated? The link and distinction needs to be noted.

Low male partner involvement in care:

Lines 469-496: The countermeasures described here are really important, in many African country settings, these ANC and delivery services are not set up to accommodate male partners attending these services with their female partners. This can be highlighted in the discussion.

Discussion:

The authors note numerous really important recommendations throughout the discussion. Are there any considerations for how to take these recommendations forward – should there be government level strategizing, call for funding to increase budgets, further research, or how can these recommendations be implemented? (the last sentence of the conclusion section highlights research on the effectiveness of the countermeasures, which addresses this).

Fear of spousal reaction and low male partner involvement in care:

Line 640: Reference this first sentence.

Strengths and limitations:

To add that only “providers” were interviewed – client perspectives could have added value to the results.

6. PLOS authors have the option to publish the peer review history of their article (what does this mean?). If published, this will include your full peer review and any attached files.

Reviewer #1: No

Reviewer #2: No

---

## [Author Response · Author response to Decision Letter 0]

12 Oct 2022

PONE-D-22-14861

Challenges and commonly used countermeasures in the implementation of lifelong antiretroviral therapy for PMTCT in Central Uganda: Health providers’ perspective.

Thank you so much for all the comments and suggestions that were raised during the review process. We thank the reviewers for all the comments. They were very helpful and have enabled us to improve our manuscript. 

We have addressed all comments in the “Point by point” response letter below. The changes are in track changes in the file labeled “Revised Manuscript with Track Changes”.

Journal Requirements:

We have followed the PLOS ONE style requirements.

“This study was funded by the Global Fund through Ministry of Health - Uganda; Grant Number: 683 UGD-708-G07-H. The contents of this article are solely the responsibility of the authors and do not necessarily represent the official views of Global Fund, Ministry of Health or Makerere University School of Public Health.”

 “This study was funded by the Global Fund through the Ministry of Health-Uganda [Grant Number: UGD-708-G07-H]. The funders had no role in study design, data collection and analysis, decision to publish, or preparation of the manuscript.”

We have removed funding-related text from the manuscript. The Funding statement should remain as stated “This study was funded by the Global Fund through the Ministry of Health-Uganda [Grant Number: UGD-708-G07-H]. The funders had no role in study design, data collection and analysis, decision to publish, or preparation of the manuscript.”

We thank you so much Academic Editor for noting issues around data availability.

There are ethical or legal restrictions to sharing our qualitative data publicly. 

Study participants were not consented to provide data access and the Higher Degrees, Research and Ethics Committee didn’t grant permission for data access. We have revised the data availability statement and provided the contact address for the Chair of the Higher Degrees, Research and Ethics Committee. 

The contact information for the ethics committee is given below. 

“The Chairperson Makerere University School of Public Health Higher Degrees, Research and Ethics Committee, P.O. Box 7072, Kampala. Telephone +256414 532207/543872/543437”

The data availability statement should now read as below:

“Due to restrictions by the Makerere University School of Public Health Higher Degrees, Research and Ethics Committee, some access restrictions apply to the data for reasons of safety and protection of study subjects and their institutions. Sensitive qualitative data was collected from study participants and they didn’t consent to open data access. However, criteria eligible researchers with interest in the data may request for anonymized data access through the Chair Higher Degrees, Research and Ethics Committee.” 

Comments to the Author

1. Is the manuscript technically sound, and do the data support the conclusions?

Reviewer #1: Yes

Reviewer #2: Yes

2. Has the statistical analysis been performed appropriately and rigorously?

Reviewer #1: N/A

Reviewer #2: N/A

3. Have the authors made all data underlying the findings in their manuscript fully available?

Reviewer #1: Yes

Reviewer #2: No

4. Is the manuscript presented in an intelligible fashion and written in standard English?

Reviewer #1: Yes

Reviewer #2: Yes

5. Review Comments to the Author

Reviewer #1: The manuscript describes the challenges and countermeasures in the implementation of lifelong ART In central Uganda. The manuscript is very well written and easy to follow and understand. There are a couple of issues when addressed will further enrich the publication

1. Introduction: It will be useful if information on transmission rates of MTCT in Uganda is included in the overview. This will enable readers put the results in perspective

We have included the information on transmission rates in the introduction section of the revised manuscript. Please see lines 87-91, on page 4.

2. 54 providers from 6 facilities and 3 districts: kindly show in a table the number and categories of providers from each health facility

A table showing characteristics including number and categories of providers has been included. This is labeled as Table 2 (Lines 235-237, on page 11). However, we have not provided a table with number and categories of providers from each facility. Providing such a table would inadvertently compromise the identity and thus confidentiality of the study participants since some categories of health providers had few interviewees per health facility.

3. Even though, the authors acknowledge that data from 2014 may be an issue but think that the results may still be relevant. I am particularly concerned because MTCT has gone through a number of transitions in the last decade. The last transition is the change to Dolutegravir with the attendant issues family planning particularly in the initial phase. In addition, viral load monitoring has become the standard of care compared to CD4 count. The implication of the results must be situated within the current context. This was not done.

We have included this information in the introduction (lines 70-73 and 78-81 on pages 3-4) and revised the limitations section to further situate our findings in the current contexts as highlighted by the reviewer. Please see changes made in lines 757-761 on page 32. 

Reviewer #2: Thank you for this interesting manuscript. Please do a final editorial review after addressing the comments.

Thank you so much for the complement. We have done an editorial review of the manuscript after addressing all the comments.

Abstract: Methods:

The authors note that there were 54 key informant interviews with providers from 6 facilities – in the manuscript, it seems that the key informants extend beyond providers, including for example expert clients (which is great for data quality) – please make this clear in abstract. Or is the term “provider” being used to include all these categories of informants?

Yes, the term “health provider” was used to mean all the categories of individuals who provided health services to the HIV positive pregnant and breastfeeding women, and their infants. We have clarified this in the abstract in lines 36 and 38 on page 2, and under the study population in the methods section (lines 135-156 on pages 6-7).

Introduction:

Historical background to ART initiation in Uganda well documented, and rationale for manuscript well presented.

Thank you.

Methods:

Study sites: Do ethics requirements allow for facilities to be named? Double check study requirements regarding identifiers and protecting confidentiality of providers. Section on ethical consideration says they were assured of confidentiality.

The ethics committee allows the health facilities to be named. The providers were de-identified by removing the names of the health facilities from the quotes to ensure confidentiality.

Study population: See comment in abstract – It is a bit confusing who the participants in this study were – the authors say 54 key informant participants were included – and then lists them (Lines 129-131) – then later, Line 137, they say 54 health providers and 57 HIV positive women were enrolled. Are the initial 54 key informant participants the same as the 54 health providers? Please make this description clearer? In the results section, it is confirmed that the 54 key informants are as they are first listed in this section. Thanks for clarifying what expert clients are. What are store assistants?

We have clarified the study population in the methods section to clearly indicate that the participants were health providers and included expert clients. We have further created two sub-sections (selection of study participants and study context) to improve clarity. Please see lines 135-161, on pages 6-7.

Stores assistants (Assistant inventory management officer) have training in procurement and supply chain management. They carry out store management, stock layout, ordering, stock taking, issue and requisition, equipment inventory, re-distribution of drugs and prepare monthly stock status reports. 

Line 133: Sentence starting “This study was part of…” – perhaps this should be the start of a new paragraph? Or clearly distinguish what is part of the study reported on in this manuscript, and what is the bigger study?

The sentence “This study was part of…” is starting on a new paragraph and we have clarified on the focus of the current manuscript in lines 162-173 on page 7. 

Lines 138-140: Once the details of the participants (all providers, or other key informants too) are clarified, make sure that this sentence outlining the aim of the manuscript is in line with this (provider’s perspective only?).

The study participants for this manuscript have been clarified. These were the 54 health providers (formal and informal). Please see lines 135-161, on pages 6-7.

Data management and analysis (and elsewhere): Where author’s initials are used, standardise if they are written with periods/full stops or not (AM vs A.M).

We have standardized the author’s initials by writing them with periods such as, A.M. throughout the manuscript. 

Results:

Figure 1 quality is not very good – quite hard to read.

We have revised figure 1 to improve its quality and ensured that it is readable.

Countermeasures – are these only described when reported on? It would be helpful to highlight this, as under most challenges they are described, and others there is no mention of how providers deal with the challenges.

To minimize the textual descriptions in the manuscript, we described the countermeasures for the challenges that were key. We included a visual lay out of the detailed challenges and countermeasures (Figure 1).

Descriptions in results – in some places these can be consolidated and summarised, as the illustrative quote gives the details – to double check that the summary point is not repetitive of what is reported in the quote (e.g. Lines 409-413: the countermeasure description could possibly be summarised as follows: “Participants reported that some clients were given additional ARV supplies to reduce visits to clinics for collection of medication” followed by the quote) – to check all.

We have checked all the results to ensure that the countermeasure descriptions are summarized where possible and that there are no repetitions in the summary point and quote. Please see lines 250-582, on pages 12-25.

Inadequacy of HIV service delivery and countermeasures:

Health worker related: Lack of relevant training:

Line 207: This quote is from a supplies officer, since the different categories of staff were interviewed, it may be helpful to specify in the quote identifier that it was a supplies officer at a hospital, not a health provider?

Supplies officers are also health providers. A supplies officer is the same as a stores’ assistant (Assistant inventory management officer). Specifying the quote identifier would compromise confidentiality since there were only two hospitals involved.

Are there any quotes/data to support the feedback of supervision and mentorship which were being done every 3 months? 

We have included a quote on supervision and mentorship. Please see lines 268-270, on page 13.

The limited training seems to indicate that not many people are exposed to this? 

Yes, this is a challenge as also highlighted that certain categories of staff were excluded from the training. Lines 255-257 on pages 11-12.

Shortage of health providers:

Line 219: Women – replace with female clients?

We have replaced “women” with female clients (lines 275 and 277 on page 13).

Line 220: Elaborate – how do long queues affect staff attitudes? Make them tired, flustered – this is linked closely to heavy workload. If too few staff, workload is more? Describe/distinguish the overlap between these points.

We have joined shortage of health providers and heavy work load so as to create a link between the two points. We have also made the points clearer. Lines 266-278 on pages 12-13.

Logistics and infrastructure related: Lack of guidelines

It seems like the guidelines do exist, it is that there are a lack of hardcopies of these guidelines? This also links to training on guidelines – if people are aware of the guidelines, they only need to refer to hardcopies for points of clarification?

This is true and we have revised the statement in the results section on page 15, lines 332-333. We have also included text in the discussion on page 29, lines 680-681.

Insufficient funds:

What should the source of these funds be? Is it budget allowance from government?

Line 303-305: Who funds these activities if there is limited funding? NGOs? Individual healthcare providers?

Most funding for PMTCT activities in Uganda is by the U.S. President’s Emergency Plan for AIDS Relief (PEPFAR) and the global fund through the Uganda Ministry of Health.

Non-utilisation of HIV services and countermeasures:

HIV related stigma: Non-disclosure of HIV positive status

Line 333: perhaps reword – “fear of marriage breakages” to “fear of negative impact on relationships” – this extends to multiple partnerships scenarios, and the example given is about a marriage break-up.

We have reworded “fear of marriage breakages” to “fear of negative impact on relationships”. Lines 396-397 on page 18.

Fear to be followed-up: Is this follow-up referring to treatment adherence follow-up, linkage to care, reminders to come for clinic visits, or what kind of follow up? It would be helpful to know what is being referred to here?

The follow-up refers to follow-up for any HIV services. This was done especially for women and/or their infants who missed scheduled clinic appointments, and those from far places. It was done through; phone calls, home visits and community out-reaches.

Unwillingness to be linked to other intra-facility service care points:

Line 367: What is EID? Make sure all acronyms in full first time.

EID is early infant diagnosis. We wrote this in full in table 1 on page 9.

Suboptimal treatment adherence and countermeasures:

ART related: Preference for herbs:

Should this section “preference for herbs” be reclassified as “preference for traditional medicines”? Is the use of herbs about using traditional medicines?

We have reclassified “preference for herbs” as “preference for traditional medicines” in lines 43,506, and 513-514.

Male spouse related: Fear of spousal reaction:

This section seems to be very closely related to the section on stigma and non-disclosure – which was because of fear of spousal reaction, resulting in non-adherence? How were these themes differentiated? The link and distinction needs to be noted.

We agree, there is a close link between non-disclosure of HIV positive results to fear of spousal reaction under HIV-related stigma. We have shifted fear of spousal reaction from lines 523-533 on page 23 to the sub section of non-disclosure on page 17 lines 391-404.

Low male partner involvement in care:

Lines 469-496: The countermeasures described here are really important, in many African country settings, these ANC and delivery services are not set up to accommodate male partners attending these services with their female partners. This can be highlighted in the discussion.

We have highlighted the importance of the countermeasures under low male involvement in African countries in the discussion section on page 31, lines 749-751.

Discussion:

The authors note numerous really important recommendations throughout the discussion. Are there any considerations for how to take these recommendations forward – should there be government level strategizing, call for funding to increase budgets, further research, or how can these recommendations be implemented? (the last sentence of the conclusion section highlights research on the effectiveness of the countermeasures, which addresses this).

We thank you for noting this.

Fear of spousal reaction and low male partner involvement in care:

Line 640: Reference this first sentence.

We referenced the first sentence. Please see page 31 line 731 (references 60 and 61). The references are

1. Mosha I, Ruben R, Kakoko D. Family planning decisions, perceptions and gender dynamics among couples in Mwanza, Tanzania: a qualitative study. BMC public health. 2013;13(1):1-13.

2. Yargawa J, Leonardi-Bee J. Male involvement and maternal health outcomes: systematic review and meta-analysis. Journal of Epidemiology and Community Health. 2015;69(6):604-12. doi: 10.1136/jech-2014-204784

Strengths and limitations:

To add that only “providers” were interviewed – client perspectives could have added value to the results.

We also interviewed clients and captured their perspectives in regards to ART uptake, adherence, HIV positive status disclosure, and HIV-related stigma were published in other papers mentioned below. We referenced these papers in our manuscript (page 7, lines 166-169 and page 26, lines 609-610)

1. Buregyeya, E., Naigino, R., Mukose, A., Makumbi, F., Esiru, G., Arinaitwe, J., Musinguzi, J. and Wanyenze, R.K., 2017. Facilitators and barriers to uptake and adherence to lifelong antiretroviral therapy among HIV infected pregnant women in Uganda: a qualitative study. BMC pregnancy and childbirth, 17(1), pp.1-9.

2. Naigino, R., Makumbi, F., Mukose, A., Buregyeya, E., Arinaitwe, J., Musinguzi, J. and Wanyenze, R.K., 2017. HIV status disclosure and associated outcomes among pregnant women enrolled in antiretroviral therapy in Uganda: a mixed methods study. Reproductive health, 14(1), pp.1-11.

---

## [Decision Letter · Decision Letter 1]

31 Oct 2022

PONE-D-22-14861R1Challenges and commonly used countermeasures in the implementation of lifelong antiretroviral therapy for PMTCT in Central Uganda: Health providers’ perspectivePLOS ONE

Dear Dr. Mukose,

Thank you for submitting your manuscript to PLOS ONE. After careful consideration, we feel that it has merit but does not fully meet PLOS ONE’s publication criteria as it currently stands. Therefore, we invite you to submit a revised version of the manuscript that addresses the points raised during the review process.

We look forward to receiving your revised manuscript.

Kind regards,

Desmond Kuupiel, PhD

Academic Editor

PLOS ONE

Journal Requirements:

Reviewers' comments:

Reviewer's Responses to Questions

**Comments to the Author**

1. If the authors have adequately addressed your comments raised in a previous round of review and you feel that this manuscript is now acceptable for publication, you may indicate that here to bypass the “Comments to the Author” section, enter your conflict of interest statement in the “Confidential to Editor” section, and submit your "Accept" recommendation.

Reviewer #1: All comments have been addressed

Reviewer #2: All comments have been addressed

2. Is the manuscript technically sound, and do the data support the conclusions?

Reviewer #1: Yes

Reviewer #2: Yes

3. Has the statistical analysis been performed appropriately and rigorously? 

Reviewer #1: N/A

Reviewer #2: N/A

4. Have the authors made all data underlying the findings in their manuscript fully available?

Reviewer #1: No

Reviewer #2: No

5. Is the manuscript presented in an intelligible fashion and written in standard English?

Reviewer #1: Yes

Reviewer #2: Yes

6. Review Comments to the Author

Reviewer #1: Authors response adequately addresses the comments raised. The manuscript is much clearer. However, the authors need to correct the error regarding Reference 2 and 3 . Otherwise, the manuscript is good to go

Reviewer #2: Thank you for addressing the review comments. I only have a few minor additional comments.

Please do a final copy edit before finalising.

Introduction:

Lines 87-91: Revision describing that MTCT of HIV should have declined due to revised treatment guidelines – the example demonstrating that it has not declined, uses data from 2017-2019, yet the guidelines were only implemented in 2020 (Line 78). Perhaps authors should find more recent data on MTCT rate in Uganda, or revise the wording of the paragraph.

Methods:

Study population: The study population selection and participation is much clearer now, thank you.

Results:

Line 229: Insert missing words: “Table 2 shows the details of the health providers who participated…”

7. PLOS authors have the option to publish the peer review history of their article (what does this mean?). If published, this will include your full peer review and any attached files.

Reviewer #1: No

Reviewer #2: No

---

## [Author Response · Author response to Decision Letter 1]

8 Nov 2022

PONE-D-22-14861R1

Challenges and commonly used countermeasures in the implementation of lifelong antiretroviral therapy for PMTCT in Central Uganda: Health providers’ perspective

Thank you so much for all the additional comments and suggestions that were raised during the review of the re-submitted manuscript. We thank the reviewers for all the comments. They have enabled us to improve our manuscript further. 

We have addressed all comments in the “Point by point” response letter below. The changes are in track changes in the file labeled “Revised Manuscript with Track Changes”.

We have included all the above items.

We have not made any changes to the financial disclosure.

We have reviewed the reference list to ensure that it is complete and correct. 

Reviewers' comments:

Reviewer's Responses to Questions

Comments to the Author

1. If the authors have adequately addressed your comments raised in a previous round of review and you feel that this manuscript is now acceptable for publication, you may indicate that here to bypass the “Comments to the Author” section, enter your conflict of interest statement in the “Confidential to Editor” section, and submit your "Accept" recommendation.

Reviewer #1: All comments have been addressed

Reviewer #2: All comments have been addressed

2. Is the manuscript technically sound, and do the data support the conclusions?

Reviewer #1: Yes

Reviewer #2: Yes

3. Has the statistical analysis been performed appropriately and rigorously?

Reviewer #1: N/A

Reviewer #2: N/A

4. Have the authors made all data underlying the findings in their manuscript fully available?

Reviewer #1: No

Reviewer #2: No

5. Is the manuscript presented in an intelligible fashion and written in standard English?

Reviewer #1: Yes

Reviewer #2: Yes

6. Review Comments to the Author

Reviewer #1: Authors response adequately addresses the comments raised. The manuscript is much clearer. However, the authors need to correct the error regarding Reference 2 and 3 Otherwise, the manuscript is good to go

Thank you. We have corrected the error in the references. They are now referenced 2 (Lines 771-772) and 4 (Lines 778-779).

Reviewer #2: Thank you for addressing the review comments. I only have a few minor additional comments.

Please do a final copy edit before finalising.

Thank you. We have edited the final copy of the manuscript.

Introduction:

Lines 87-91: Revision describing that MTCT of HIV should have declined due to revised treatment guidelines – the example demonstrating that it has not declined, uses data from 2017-2019, yet the guidelines were only implemented in 2020 (Line 78). Perhaps authors should find more recent data on MTCT rate in Uganda, or revise the wording of the paragraph.

We have revised the wording and included more recent data as well (Lines 95-102).

Methods:

Study population: The study population selection and participation is much clearer now, thank you.

Thank you very much.

Results:

Line 229: Insert missing words: “Table 2 shows the details of the health providers who participated…”

We have revised the wording of the statements (Lines 221 and 227).

7. PLOS authors have the option to publish the peer review history of their article (what does this mean?). If published, this will include your full peer review and any attached files.

Do you want your identity to be public for this peer review? For information about this choice, including consent withdrawal, please see our Privacy Policy.

Reviewer #1: No

Reviewer #2: No

---

## [Editor Report · Decision Letter 2]

11 Jan 2023

Challenges and commonly used countermeasures in the implementation of lifelong antiretroviral therapy for PMTCT in Central Uganda: Health providers’ perspective

PONE-D-22-14861R2

Dear Dr. Mukose,

We’re pleased to inform you that your manuscript has been judged scientifically suitable for publication and will be formally accepted for publication once it meets all outstanding technical requirements.

Kind regards,

Desmond Kuupiel, PhD

Academic Editor

PLOS ONE
---

## [Editor Report · Acceptance letter]

12 Jan 2023

PONE-D-22-14861R2 

Challenges and commonly used countermeasures in the implementation of lifelong antiretroviral therapy for PMTCT in Central Uganda: Health providers’ perspective 

Dear Dr. Mukose:

I'm pleased to inform you that your manuscript has been deemed suitable for publication in PLOS ONE. Congratulations! Your manuscript is now with our production department. 

Kind regards, 

on behalf of

Dr. Desmond Kuupiel 

Academic Editor

PLOS ONE